# Semi-supervised Multitask Learning for Sequence Labeling

## Abstract

We propose a sequence labeling framework with a secondary training objective, learning to predict surrounding words for every word in the dataset. This language modeling objective incentivises the framework to learn general-purpose patterns of semantic and syntactic composition, which are also useful for improving accuracy on different sequence labeling tasks. The architecture was evaluated on 8 datasets, covering the tasks of error detection in learner texts, named entity recognition, chunking and POS-tagging. The novel language modeling objective provided consistent performance improvements on every benchmark, without requiring any additional annotated or unannotated data.

## 1 Introduction

Accurate and efficient sequence labeling models have a wide range of applications, including named entity recognition (NER), part-of-speech (POS) tagging, error detection and shallow parsing. Specialised approaches to sequence labeling often include extensive feature engineering, such as integrated gazetteers, capitalisation features, morphological information and POS tags. However, recent work has shown that neural network architectures are able to achieve comparable or improved performance, while automatically discovering useful features for a specific task and only requiring a sequence of tokens as input (Collobert et al., 2011; Irsoy and Cardie, 2014; Lample et al., 2016).

This feature discovery is usually driven by an objective function based on predicting the annotated labels for each word, without much incentive to learn more general language features from the available text. In many sequence labeling tasks, the relevant labels in the dataset are very sparse and most of the words contribute very little to the training process. For example, in the CoNLL 2003 NER dataset (Tjong Kim Sang and De Meulder, 2003) only 17% of the tokens represent an entity. This ratio is even lower for error detection, with only 14% of all tokens being annotated as an error in the FCE dataset (Yannakoudakis et al., 2011). The sequence labeling models are able to learn this bias in the label distribution without obtaining much additional information from words that have the majority label (O for outside of an entity; C for correct word). Therefore, we propose an additional training objective which allows the models to make more extensive use of the available data.

The task of language modeling offers an easily accessible objective – learning to predict the next word in the sequence requires only plain text as input, without relying on any particular annotation. Neural language modeling architectures also have many similarities to common sequence labeling frameworks: words are first mapped to distributed embeddings, followed by a recurrent neural network (RNN) module for composing word sequences into an informative context representation (Mikolov et al., 2010; Graves et al., 2013; Chelba et al., 2013). Compared to any sequence labeling dataset, the task of language modeling has a considerably larger and more varied set of possible options to predict, making better use of each available word and encouraging the model to learn more general language features for accurate composition.

In this paper, we propose a neural sequence labeling architecture that is also optimised as a language model, predicting surrounding words in the dataset in addition to assigning labels to each token. Specific sections of the network are op-

timised as a forward- or backward-moving language model, while the label predictions are performed using context from both directions. This secondary unsupervised objective encourages the framework to learn richer features for semantic composition without requiring additional training data. We evaluate the sequence labeling model on 8 datasets from the fields of NER, POS-tagging, chunking and error detection in learner texts. Our experiments show that by including the unsupervised objective into the training process, the sequence labeling model achieves consistent performance improvements on all the benchmarks. This multitask training framework gives the largest improvements on error detection datasets, where the system also achieves new state-of-the-art results.

## 2 Neural Sequence Labeling

We use the neural network model by Rei et al. (2016) as the baseline architecture for our sequence labeling experiments. The model takes as input one sentence, separated into tokens, and assigns a label to every token using a bidirectional LSTM.

The input tokens are first mapped to a sequence of distributed word embeddings $[x_1, x_2, x_3, ..., x_T]$. Two LSTM (Hochreiter and Schmidhuber, 1997) components, moving in opposite directions through the sentence, are then used for constructing context-dependent representations for every word. Each LSTM takes as input the hidden state from the previous time step, along with the word embedding from the current step, and outputs a new hidden state. The hidden representations from both directions are concatenated, in order to obtain a context-specific representation for each word that is conditioned on the whole sentence in both directions:

$$\overrightarrow{h_t} = LSTM(x_t, \overrightarrow{h_{t-1}}) \quad (1)$$

$$\overleftarrow{h_t} = LSTM(x_t, \overleftarrow{h_{t+1}}) \quad (2)$$

$$h_t = [\overrightarrow{h_t}; \overleftarrow{h_t}] \quad (3)$$

Next, the concatenated representation is passed through a feedforward layer, mapping the components into a joint space and allowing the model to learn features based on both context directions:

$$d_t = tanh(W_d h_t) \quad (4)$$

where $W_d$ is a weight matrix and $tanh$ is used as the non-linear activation function.

In order to predict a label for each token, we use either a softmax or CRF output architecture. For softmax, the model directly predicts a normalised distribution over all possible labels for every word, conditioned on the vector $d_t$:

$$P(y_t|d_t) = softmax(W_o d_t)$$
$$= \frac{e^{W_{o,k} d_t}}{\sum_{\tilde{k} \in K} e^{W_{o,\tilde{k}} d_t}} \quad (5)$$

where $K$ is the set of all possible labels, and $W_{o,k}$ is the $k$-th row of output weight matrix $W_o$. The model is optimised by minimising categorical crossentropy, which is equivalent to minimising the negative log-probability of the correct labels:

$$E = -\sum_{t=1}^{T} log(P(y_t|d_t)) \quad (6)$$

While this architecture returns predictions based on all words in the input, the labels are still predicted independently. For some tasks, such as named entity recognition with a BIO[1] scheme, there are strong dependencies between subsequent labels and it can be beneficial to explicitly model these connections. The output of the architecture can be modified to include a Conditional Random Field (CRF, Lafferty et al. (2001)), which allows the network to look for the most optimal path through all possible label sequences (Huang et al., 2015). The model is then optimised by maximising the score for the correct label sequence, while minimising the scores for all other sequences:

$$E = -s(y) + log \sum_{\tilde{y} \in \widetilde{Y}} e^{s(\tilde{y})} \quad (7)$$

where $s(y)$ is the score for a given sequence $y$ and $Y$ is the set of all possible label sequences.

We also make use of the character-level component described by Rei et al. (2016), which builds an alternative representation for each word. The individual characters of a word are mapped to character embeddings and passed through a bidirectional LSTM. The last hidden states from both direction are concatenated and passed through a

---

[1]Each NER entity has sub-tags for Beginning, Inside and Outside

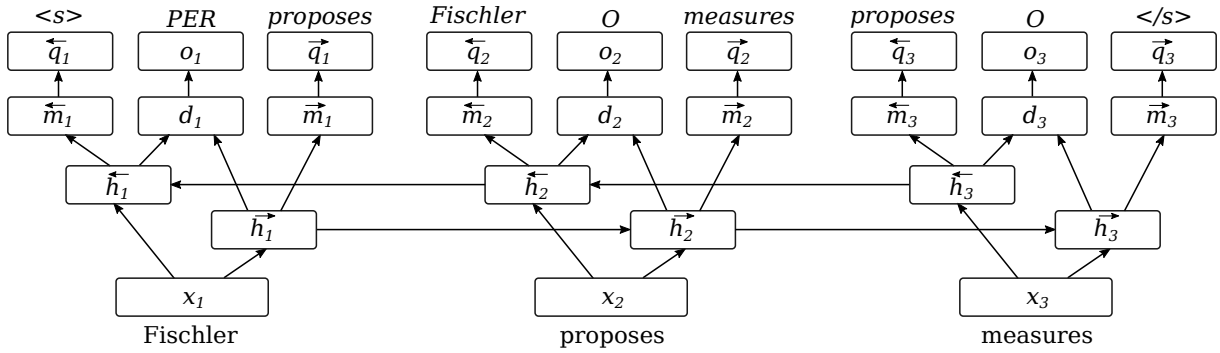

Figure 1: The unfolded network structure for a sequence labeling model with an additional language modeling objective, performing NER on the sentence *"Fischler proposes measures"*. The input tokens are shown at the bottom, the expected output labels are at the top. Arrows above variables indicate the directionality of the component (forward or backward).

nonlinear layer. The resulting vector representation is combined with a regular word embedding using a dynamic weighting mechanism that adaptively controls the balance between word-level and character-level features. This framework allows the model to learn character-based patterns and handle previously unseen words, while still taking full advantage of the word embeddings.

## 3 Language Modeling Objective

The sequence labeling model in Section 2 is only optimised based on the correct labels. While each token in the input does have a desired label, many of these contribute very little to the training process. For example, in the CoNLL 2003 NER dataset (Tjong Kim Sang and De Meulder, 2003) there are only 8 possible labels and 83% of the tokens have the label O, indicating that no named entity is detected. This ratio is even higher for error detection, with 86% of all tokens containing no errors in the FCE dataset (Yannakoudakis et al., 2011). The sequence labeling models are able to learn this bias in the label distribution without obtaining much additional information from the majority labels. Therefore, we propose an additional objective which would allow the models to make full use of the training data.

In addition to learning to predict labels for each word, we propose optimising specific sections of the architecture as language models. The task of predicting the next word will require the model to learn more general patterns of semantic and syntactic composition, which can then be reused in order to predict individual labels more accurately. This objective is also generalisable to any

sequence labeling task and dataset, as it requires no additional annotated training data.

A straightforward modification of the sequence labeling model would add a second parallel output layer for each token, optimising it to predict the next word. However, the model has access to the full context on each side of the target token, and predicting information that is already explicit in the input would not incentivise the model to learn about composition and semantics. Therefore, we must design the loss objective so that only sections of the model that have not yet observed the next word are optimised to perform the prediction. We solve this by predicting the next word in the sequence only based on the hidden representation $\overrightarrow{h_t}$ from the forward-moving LSTM. Similarly, the previous word in the sequence is predicted based on $\overleftarrow{h_t}$ from the backward-moving LSTM. This architecture avoids the problem of giving the correct answer as an input to the language modeling component, while the full framework is still optimised to predict labels based on the whole sentence.

First, the hidden representations from forward- and backward-LSTMs are mapped to a new space using a non-linear layer:

$$\overrightarrow{m_t} = tanh(\overrightarrow{W}_m \overrightarrow{h_t}) \qquad (8)$$

$$\overleftarrow{m_t} = tanh(\overleftarrow{W}_m \overleftarrow{h_t}) \qquad (9)$$

where $\overrightarrow{W}_m$ and $\overleftarrow{W}_m$ are weight matrices. This separate transformation learns to extract features that are specific to language modeling, while the LSTM is optimised for both objectives. We also use the opportunity to map the representation to a smaller size – since language modeling is not the

main goal, we restrict the number of parameters available for this component, forcing the model to generalise more using fewer resources.

These representations are then used to predict the preceding and following word, by using a softmax layer with the size of the full vocabulary:

$$P(w_{t+1}|\overrightarrow{m_t}) = softmax(\overrightarrow{W}_q\overrightarrow{m_t}) \qquad (10)$$

$$P(w_{t-1}|\overleftarrow{m_t}) = softmax(\overleftarrow{W}_q\overleftarrow{m_t}) \qquad (11)$$

The objective function for both components is then constructed as a regular language modeling objective, by calculating the negative log-likelihood of the next word in the sequence:

$$\overrightarrow{E} = -\sum_{t=1}^{T-1} log(P(w_{t+1}|\overrightarrow{m_t})) \qquad (12)$$

$$\overleftarrow{E} = -\sum_{t=2}^{T} log(P(w_{t-1}|\overleftarrow{m_t})) \qquad (13)$$

Finally, these additional objectives are combined with the training objective $E$ from either Equation 6 or 7, resulting in a new cost function $\widetilde{E}$ for the sequence labeling model:

$$\widetilde{E} = E + \gamma(\overrightarrow{E} + \overleftarrow{E}) \qquad (14)$$

where $\gamma$ is a parameter that is used to control the importance of the language modeling objective in comparison to the sequence labeling objective.

Figure 1 shows a diagram of the unfolded neural architecture, when performing NER on a short sentence with 3 words. At each token position, the network is optimised to predict the previous word, the current label, and the next word in the sequence. The added language modeling objective encourages the system to learn richer feature representations that are then reused for sequence labeling. For example, $\overrightarrow{h_1}$ is optimised to predict *proposes* as the next word, indicating that the current word is a subject, possibly a named entity. In addition, $\overleftarrow{h_2}$ is optimised to predict *Fischler* as the previous word and these features are used as input to predict the *PER* tag at $o_1$.

The proposed architecture introduces 4 additional parameter matrices that are optimised during training: $\overrightarrow{W}_m$, $\overleftarrow{W}_m$, $\overrightarrow{W}_q$, and $\overleftarrow{W}_q$. However, the computational complexity and resource requirements of this model during sequence labeling are equal to the baseline from Section 2, since the language modeling components are ignored during testing and these additional weight matrices are not used. While our implementation uses a basic softmax as the output layer for the language modeling components, the efficiency during training could be further improved by integrating noise-contrastive estimation (NCE, Mnih and Teh (2012)) or hierarchical softmax (Morin and Bengio, 2005).

## 4 Evaluation Setup

The proposed architecture was evaluated on 8 different sequence labeling datasets, covering the tasks of error detection, NER, chunking, and POS-tagging. The word embeddings in the model were initialised with publicly available pretrained vectors, created using word2vec (Mikolov et al., 2013). For general-domain datasets we used 300-dimensional embeddings trained on Google News.[2] For biomedical datasets, the word embeddings were initialised with 200-dimensional vectors trained on PubMed and PMC.[3]

The neural network framework was implemented using Theano (Al-Rfou et al., 2016) and we make the code publicly available online.[4] For most of the hyperparameters, we follow the settings by Rei et al. (2016) in order to facilitate direct comparison with previous work. The LSTM hidden layers are set to size 200 in each direction for both word- and character-level components. All digits in the text were replaced with the character 0; any words that occurred only once in the training data were replaced by an OOV token. Sentences were grouped into batches of size 64 and parameters were optimised using AdaDelta (Zeiler, 2012) with default learning rate 1.0. Training was stopped when performance on the development set had not improved for 7 epochs. Performance on the development set was also used to select the best model, which was then evaluated on the test set. In order to avoid any outlier results due to randomness in the model initialisation, each configuration was trained with 10 different random seeds and the averaged results are presented in this paper. We use previously established splits for training, development and testing

---

[2]https://code.google.com/archive/p/word2vec/
[3]http://bio.nlplab.org/
[4]URL removed for anonymisation

|  | FCE DEV | FCE TEST | | | CoNLL-14 TEST1 | | | CoNLL-14 TEST2 | | |
|---|---|---|---|---|---|---|---|---|---|---|
|  | $F_{0.5}$ | P | R | $F_{0.5}$ | P | R | $F_{0.5}$ | P | R | $F_{0.5}$ |
| Baseline | 48.78 | 55.38 | 25.34 | 44.56 | 15.65 | 16.80 | 15.80 | 25.22 | 19.25 | 23.62 |
| + dropout | 48.68 | 54.11 | 23.33 | 42.65 | 14.29 | 17.13 | 14.71 | 22.79 | 19.42 | 21.91 |
| + LMcost | **53.17** | **58.88** | **28.92** | **48.48** | **17.68** | **19.07** | **17.86** | **27.62** | **21.18** | **25.88** |

Table 1: Precision, Recall and $F_{0.5}$ score of alternative sequence labeling architectures on error detection datasets. Dropout and LMcost modifications are added incrementally to the baseline.

on each of these datasets.

Based on development experiments, we found that error detection was the only task that did not benefit from having a CRF module at the output layer – since the labels are very sparse and the dataset contains only 2 possible labels, explicitly modeling state transitions does not improve performance on this task. Therefore, we use a softmax output for error detection experiments and CRF on all other datasets.

The publicly available sequence labeling system by Rei et al. (2016) is used as the baseline.[5] During development we found that applying dropout (Srivastava et al., 2014) on word embeddings improves performance on nearly all datasets, compared to this baseline. Therefore, element-wise dropout was applied to each of the input embeddings with probability $0.5$ during training, and the weights were multiplied by $0.5$ during testing. In order to separate the effects of this modification from the newly proposed optimisation method, we report results for three different systems: 1) the publicly available baseline, 2) applying dropout on top of the baseline system, and 3) applying both dropout and the novel multitask objective from Section 3.

Based on development experiments we set the value of $\gamma$, which controls the importance of the language modeling objective, to $0.1$ for all experiments throughout training. Since context prediction is not part of the main evaluation of sequence labeling systems, we expected the additional objective to mostly benefit early stages of training, whereas the model would later need to specialise only towards assigning labels. Therefore, we also performed experiments on the development data where the value of $\gamma$ was gradually decreased, but found that a small static value performed comparably well or even better. These experiments indicate that the language modeling objective helps

the network learn general-purpose features that are useful for sequence labeling even in the later stages of training.

## 5 Error Detection

We first evaluate the sequence labeling architectures on the task of error detection – given a sentence written by a language learner, the system needs to detect which tokens have been manually tagged by examiners as being an error. As the first benchmark, we use the publicly released First Certificate in English (FCE) dataset, containing 33,673 manually annotated sentences. The texts were written by learners during language examinations in response to prompts eliciting free-text answers and assessing mastery of the upper-intermediate proficiency level. In addition, we evaluate on the CoNLL 2014 shared task dataset (Ng et al., 2014), which has been converted to an error detection task. This contains 1,312 sentences, written by higher-proficiency learners on more technical topics. They have been manually corrected by two separate annotators, and we report results on each of these annotations. For both datasets we use the FCE training set for model optimisation and results on the CoNLL-14 dataset indicate out-of-domain performance. Rei and Yannakoudakis (2016) present results on these datasets using the same setup, along with evaluating the top shared task submissions on the task of error detection. As the main evaluation metric, we use the $F_{0.5}$ measure, which is consistent with previous work and was also adopted by the CoNLL-14 shared task.

Table 1 contains results for the three different sequence labeling architectures on the error detection datasets. The baseline results are comparable to the previous best results on each of these benchmarks. We found that including the dropout actually decreases performance in the setting of error detection, which is likely due to the rela-

---
[5] https://github.com/marekrei/sequence-labeler

|  | CoNLL-00 | | CoNLL-03 | | CHEMDNER | | JNLPBA | |
|---|---|---|---|---|---|---|---|---|
|  | DEV | TEST | DEV | TEST | DEV | TEST | DEV | TEST |
| Baseline | 92.92 | 92.67 | 90.85 | 85.63 | 83.63 | 84.51 | 77.13 | 72.79 |
| + dropout | 93.40 | 93.15 | 91.14 | 86.00 | 84.78 | 85.67 | 77.61 | 73.16 |
| + LMcost | **94.22** | **93.88** | **91.48** | **86.26** | **85.45** | **86.27** | **78.51** | **73.83** |

Table 2: Performance of alternative sequence labeling architectures on NER and chunking datasets, measured using CoNLL standard entity-level $F_1$ score.

tively small amount of error examples available in the dataset – it is better for the model to memorise them without introducing additional noise in the form of dropout. However, we did verify that dropout indeed gives an improvement in combination with the novel language modeling objective. Because the model is receiving additional information at every token, dropout is no longer obscuring the limited training data but instead helps with generalisation.

The bottom row shows the performance of the language modeling objective when added on top of the baseline model, along with dropout on word embeddings. This architecture outperforms the baseline on all benchmarks, increasing both precision and recall, and giving a 3.9% absolute improvement on the FCE test set. These are also the new state-of-the-art results for error detection on both FCE and CoNLL-14 datasets. Error detection is the task where introducing the additional cost objective gave the largest improvement in performance, for a few reasons:

1. This task has very sparse labels in the datasets, with error tokens very infrequent and far apart. Without the language modeling objective, the network has very little use for all the available words that contain no errors.

2. There are only two possible labels, correct and incorrect, which likely makes it more difficult for the model to learn feature detectors for many different error types. Language modeling uses a much larger number of possible labels, giving a more varied training signal.

3. Finally, the task of error detection is directly related to language modeling. By learning a better model of the overall text in the training corpus, the system can more easily detect any irregularities.

We also analysed the performance of the different architectures during training. Figure 2 shows the $F_{0.5}$ score on the development set for each model after every epoch over the training data. The baseline model peaks quickly, followed by a gradual drop in performance, which is likely due to overfitting on the available data. Dropout provides an effective regularisation method, slowing down the initial performance but preventing the model from overfitting. The added language modeling objective provides a substantial improvement – the system outperforms other configurations already in the early stages of training and the results are also sustained in the later epochs.

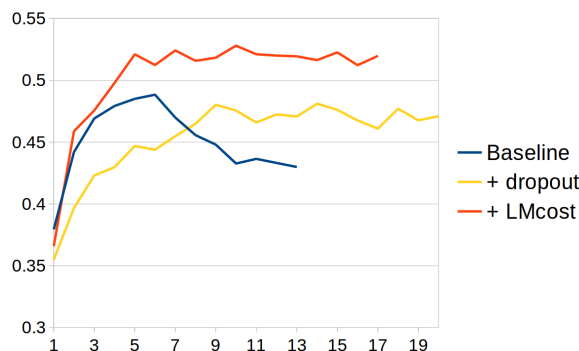

Figure 2: $F_{0.5}$ score on the FCE development set after each training epoch.

## 6 NER and Chunking

In the next experiments we evaluate the language modeling objective on named entity recognition and chunking. For general-domain NER, we use the English section of the CoNLL 2003 corpus (Tjong Kim Sang and De Meulder, 2003), containing news stories from the Reuters Corpus. We also report results on two biomedical NER datasets: The BioCreative IV Chemical and Drug corpus (CHEMDNER, Krallinger et al. (2015)) of 10,000 abstracts, annotated for mentions of chemical and drug names, and the JNLPBA corpus (Kim et al.,

2004) of 2,404 abstracts annotated for mentions of different cells and proteins. Finally, we use the CoNLL 2000 dataset (Tjong Kim Sang and Buchholz, 2000), created from the Wall Street Journal Sections 15-18 and 20 from the Penn Treebank, for evaluating sequence labeling on the task of chunking. The standard CoNLL entity-level $F_1$ score is used as the main evaluation metric.

Compared to error detection corpora, the labels are more balanced in these datasets. However, majority labels still exist: roughly 83% of the tokens in the NER datasets are tagged as "O", indicating that the word is not an entity, and the NP label covers 53% of tokens in the chunking data.

Table 2 contains results for evaluating the different architectures on NER and chunking. On these tasks, the application of dropout provides a consistent improvement – applying some variance onto the input embeddings results in more robust models for NER and chunking. The addition of the language modeling objective consistently further improves performance on all benchmarks.

While these results are comparable to the respective state-of-the-art results on most datasets, we did not fine-tune hyperparameters for any specific task, instead providing a controlled analysis of the language modeling objective in different settings. For JNLPBA, the system achieves 73.83% compared to 72.55% by Zhou and Su (2004) and 72.70% by Rei et al. (2016). On CoNLL-03, Lample et al. (2016) achieve a considerably higher result of 90.94%, possibly due to their use of specialised word embeddings and a custom version of LSTM. However, our system does outperform a similar architecture by Huang et al. (2015), achieving 86.26% compared to 84.26% $F_1$ score on the CoNLL-03 dataset.

Figure 3 shows $F_1$ on the CHEMDNER development set after every training epoch. Without dropout, performance peaks quickly and then trails off as the system overfits on the training set. Using dropout, the best performance is sustained throughout training and even slightly improved. Finally, adding the language modeling objective on top of dropout allows the system to consistently outperform the other architectures.

## 7 POS tagging

We also evaluated the language modeling training objective on two POS-tagging datasets. The Penn Treebank POS-tag corpus (Marcus et al., 1993)

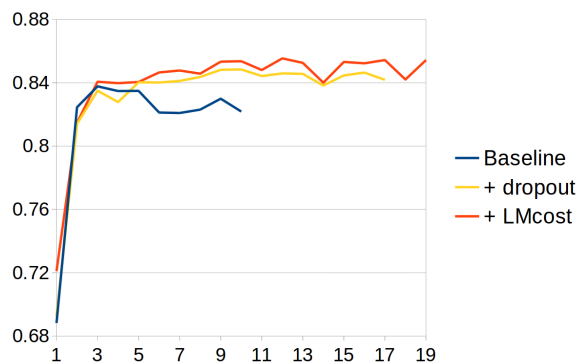

Figure 3: Entity-level $F_1$ score on the CHEMD-NER development set after each training epoch.

contains texts from the Wall Street Journal and has been annotated with 48 different part-of-speech tags. In addition, we use the POS-annotated subset of the GENIA corpus (Ohta et al., 2002) containing 2,000 biomedical PubMed abstracts. Following Tsuruoka et al. (2005), we use the same 210-document test set.

These datasets are somewhat more balanced in terms of label distributions, compared to error detection and NER, as no single label covers over 50% of the tokens. POS-tagging also offers a large variance of unique labels, with 48 labels in PTB and 42 in GENIA, and this can provide useful information to the models during training. The baseline performance on these datasets is also close to the upper bound, therefore we expect the language modeling objective to not provide much additional benefit.

The results of different sequence labeling architectures on POS-tagging can be seen in Table 3 and accuracy on the development set is shown in Figure 4. While the performance improvements are small, they are consistent across both domains and datasets. Application of dropout again provides a more robust model, and the language modeling cost improves the performance further. Even though the labels already offer a varied training objective, learning to predict the surrounding words at the same time has provided the model with additional general-purpose features.

## 8 Related Work

Our work builds on previous research exploring multi-task learning in the context of different sequence labeling tasks. The idea of multi-task learning was described by Caruana (1998) and has since been extended to many language process-

|  | GENIA-POS | | PTB-POS | |
|---|---|---|---|---|
|  | DEV | TEST | DEV | TEST |
| Baseline | 98.69 | 98.61 | 97.23 | 97.24 |
| + dropout | 98.79 | 98.71 | 97.36 | 97.30 |
| + LMcost | **98.89** | **98.81** | **97.48** | **97.43** |

Table 3: Accuracy of different sequence labeling architectures on POS-tagging datasets.

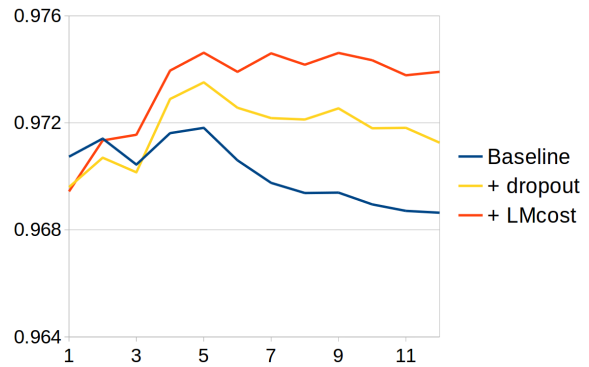

Figure 4: Token-level accuracy on the PTB-POS development set after each training epoch.

ing tasks using neural networks. For example, Collobert and Weston (2008) proposed a multi-task framework using weight-sharing between networks that are optimised for different supervised tasks.

Cheng et al. (2015) described a system for detecting out-of-vocabulary names by also predicting the next word in the sequence. While they use a regular recurrent architecture, we propose a language modeling objective that can be integrated with a bidirectional network, making it applicable to existing state-of-the-art sequence labeling frameworks.

Plank et al. (2016) described a related architecture for POS-tagging, predicting the frequency of each word together with the part-of-speech, and showed that this can improve tagging accuracy on low-frequency words. While predicting word frequency is useful for POS-tagging, language modeling provides a more general training signal, allowing us to apply the model to many different sequence labeling tasks. Our framework also achieves higher accuracy on the PTB-POS dataset (97.43% vs 97.22%).

## 9 Conclusion

We proposed a novel sequence labeling framework with a secondary objective – learning to predict surrounding words for each word in the dataset. One half of a bidirectional LSTM is trained as a forward-moving language model, whereas the other half is trained as a backward-moving language model. At the same time, both of these are also combined, in order to predict the most probable label for each word. This modification can be applied to several common sequence labeling architectures and requires no additional annotated or unannotated data.

The objective of learning to predict surrounding words provides an additional source of information during training. The model is incentivised to discover useful features in order to learn the language distribution and composition patterns in the training data. While language modeling is not the main goal of the system, this additional training objective leads to more accurate sequence labeling models on several different tasks.

The architecture was evaluated on 8 different datasets, covering the tasks of error detection in learner texts, named entity recognition, chunking and POS-tagging. We found that the additional language modeling objective provided consistent performance improvements on every benchmark. The largest benefit from the new architecture was observed on the task of error detection in learner writing. The label distribution in the original dataset is very sparse and unbalanced, making it a difficult task for the model to learn. The added language modeling objective allowed the system to take better advantage of the available training data, leading to 3.9% absolute improvement over the previous state-of-the-art. The language modeling objective also provided consistent improvements on other sequence labeling tasks, such as named entity recognition, chunking and POS-tagging.

Future work could investigate the extension of this architecture to additional unannotated resources. Learning generalisable language features from large amounts of unlabeled in-domain text could provide sequence labeling models with additional benefit. While it is common to pre-train word embeddings on large-scale unannotated corpora, only minimal work has looked at useful methods for pre-training or co-training more advanced compositional modules.

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
