# Peer review of "Semi-supervised Multitask Learning for Sequence Labeling"

_ACL 2017 — decision unknown_

[Official Review · Reviewer 1 · rating 3 · confidence 4]
soundness 5 · originality 5 · clarity 4 · impact 3 · substance 2 · appropriateness 5 · meaningful comparison 3 · presentation format Oral Presentation

The paper proposes an approach to sequence labeling with multitask learning,
where language modeling is uses as the auxiliary objective. Thus, a
bidirectional neural network architecture learns to predict the output labels
as well as to predict the previous or next word in the sentence. The joint
objectives lead to improvements over the baselines in grammatical error
detection, chunking, NER, and POS tagging.

- Strengths:

The contribution is quite well-written and easy to follow for the most part.
The model is exposed in sufficient detail, and the experiments are thorough
within the defined framework. The benefits of introducing an auxiliary
objective are nicely exposed.

- Weaknesses:

The paper shows very limited awareness of the related work, which is extensive
across the tasks that the experiments highlight. Tables 1-3 only show the three
systems proposed by the contribution (Baseline, +dropout, and +LMcost), while
some very limited comparisons are sketched textually.

A contribution claiming novelty and advancements over the previous state of the
art should document these improvements properly: at least by reporting the
relevant scores together with the novel ones, and ideally through replication.
The datasets used in the experiments are all freely available, the previous
results well-documented, and the previous systems are for the most part
publicly available.

In my view, for a long paper, it is a big flaw not to treat the previous work
more carefully.

In that sense, I find this sentence particularly troublesome: "The baseline
results are comparable to the previous best results on each of these
benchmarks." The reader is here led to believe that the baseline system somehow
subsumes all the previous contributions, which is shady on first read, and
factually incorrect after a quick lookup in related work.

The paper states "new state-of-the-art results for error detection on both FCE
and CoNLL-14 datasets". Looking into the CoNLL 2014 shared task report, it is
not straightforward to discern whether the
latter part of the claim does holds true, also as per Rei and Yannakoudakis'
(2016) paper. The paper should support the claim by inclusion/replication of
the related work.

- General Discussion:

The POS tagging is left as more of an afterthought. The comparison to Plank et
al. (2016) is at least partly unfair as they test across multiple languages in
the Universal Dependencies realm, showing top-level performance across language
families, which I for one believe to be far more relevant than WSJ
benchmarking. How does the proposed system scale up/down to multiple languages,
low-resource languages with limited training data, etc.? The paper leaves a lot
to ask for in that dimension to further substantiate its claims.

I like the idea of including language modeling as an auxiliary task. I like the
architecture, and sections 1-4 in general. In my view, there is a big gap
between those sections and the ones describing the experiments (5-8).

I suggest that this nice idea should be further fleshed out before publication.
The rework should include at least a more fair treatment of related work, if
not replication, and at least a reflection on multilinguality. The data and the
systems are all there, as signs of the field's growing maturity. The paper
should in my view partake in reflecting this maturity, and not step away from
it. In faith that these improvements can be implemented before the publication
deadline, I vote borderline.

[Official Review · Reviewer 2 · rating 4 · confidence 4]
soundness 5 · originality 5 · clarity 5 · impact 3 · substance 3 · appropriateness 5 · meaningful comparison 3 · presentation format Poster

- Strengths: The article is well written; what was done is clear and
straightforward. Given how simple the contribution is, the gains are
substantial, at least in the error correction task.

- Weaknesses: The novelty is fairly limited (essentially, another permutation
of tasks in multitask learning), and only one way of combining the tasks is
explored. E.g., it would have been interesting to see if pre-training is
significantly worse than joint training; one could initialize the weights from
an existing RNN LM trained on unlabeled data; etc.

- General Discussion: I was hesitating between a 3 and a 4. While the
experiments are quite reasonable and the combinations of tasks sometimes new,
there's quite a bit of work on multitask learning in RNNs (much of it already
cited), so it's hard to get excited about this work. I nevertheless recommend
acceptance because the experimental results may be useful to others.

- Post-rebuttal: I've read the rebuttal and it didn't change my opinion of the
paper.